# Comparisons of Radiofrequency Ablation, Microwave Ablation, and Irreversible Electroporation by Using Propensity Score Analysis for Early Stage Hepatocellular Carcinoma

**DOI:** 10.3390/cancers15030732

**Published:** 2023-01-25

**Authors:** Takuya Wada, Katsutoshi Sugimoto, Kentaro Sakamaki, Hiroshi Takahashi, Tatsuya Kakegawa, Yusuke Tomita, Masakazu Abe, Yu Yoshimasu, Hirohito Takeuchi, Takao Itoi

**Affiliations:** 1Department of Gastroenterology and Hepatology, Tokyo Medical University, 6-7-1 Nishishinjuku, Shinjuku-ku, Tokyo 160-0023, Japan; 2Center for Data Science, Yokohama City University, 22-2 Seto, Kanazawa-ku, Yokohama-shi 236-0027, Kanagawa, Japan

**Keywords:** hepatocellular carcinoma, radiofrequency ablation, microwave ablation, irreversible electroporation

## Abstract

**Simple Summary:**

This single-center retrospective study aimed to compare the therapeutic and safety outcomes of radiofrequency ablation (RFA), microwave ablation (MWA), and irreversible electroporation (IRE) in the treatment of early stage hepatocellular carcinoma (HCC) using propensity score-matched analysis to reduce selection bias. A significant difference in 2-year local tumor progression (LTP) rates between the IRE and RFA groups (IRE, 0.0% vs. RFA, 45.0%; *p* = 0.005) was found. There was no significant difference in 2-year LTP rates between the IRE and MWA groups (IRE, 0.0% vs. MWA, 25.0%; *p* = 0.103) as well as between the RFA and MWA groups (RFA, 18.2% vs. MWA, 20.6%; *p* = 0.586). IRE provides better local tumor control than RFA as a first-line therapeutic option for small perivascular HCC.

**Abstract:**

Background: Despite the diversity of thermal ablations, such as radiofrequency ablation (RFA) and microwave ablation (MWA), and non-thermal ablation, such as irreversible electroporation (IRE) cross-comparisons of multiple ablative modalities for hepatocellular carcinoma (HCC) treatment remain scarce. Thus, we investigated the therapeutic outcomes of different three ablation modalities in the treatment of early stage HCC. Methods: A total of 322 consecutive patients with 366 HCCs (mean tumor size ± standard deviation: 1.7 ± 0.9 cm) who underwent RFA (n = 216, 59.0%), MWA (n = 91, 28.3%), or IRE (n = 15, 4.7%) were included. Local tumor progression (LTP) rates for LTP were compared among the three modalities. Propensity score-matched analysis was used to reduce selection bias. Results: A significant difference in 2-year LTP rates between the IRE and RFA groups (IRE, 0.0% vs. RFA, 45.0%; *p* = 0.005) was found. There was no significant difference in 2-year LTP rates between the IRE and MWA groups (IRE, 0.0% vs. MWA, 25.0%; *p* = 0.103) as well as between the RFA and MWA groups (RFA, 18.2% vs. MWA, 20.6%; *p* = 0.586). Conclusion: IRE provides better local tumor control than RFA as a first-line therapeutic option for small perivascular HCC.

## 1. Introduction

Thermal ablation therapy, such as radiofrequency ablation (RFA), is the most commonly used technology that has been investigated for the treatment of early stage hepatocellular carcinoma (HCC) that is unsuitable for surgical resection. A recent randomized control trial showed that progression-free survival rates for HCC < 3 cm in diameter and less than three HCCs treated using RFA or surgical resection are comparable [1]. Accordingly, RFA is indicated for patients with up to three HCCs that are < 3 cm in diameter and Child-Pugh class A or B liver function [2,3].

In recent years, second-generation microwave ablation (MWA) has been used as another thermal ablation technology for the treatment of HCC [4]. Although there has been less conclusive evidence on the use of MWA in HCC than RFA use, the major advantage of MWA is that it can provide large and spherical ablation areas in a single ablation session, which may have less heat-sink effect compared with RFA, resulting in a lower local tumor progression (LTP) rate [5].

Moreover, non-thermal ablation therapy, such as irreversible electroporation (IRE), has also been used for the treatment of HCC [6]. Although IRE irreversibly injures the membranes of all cells in the target tissue, the preservation of extracellular macromolecules and constitutive connective tissue components spares the structural integrity of the tissue. This characteristic theoretically makes IRE an attractive therapy for tumors in the vicinity of vital structures, such as large blood vessels, intestines, and biliary tracts [7].

Despite the diversity of thermal ablations, such as RFA and MWA, and non-thermal ablation, such as IRE, cross-comparisons of multiple ablative modalities for HCC treatment remain scarce due to their different costs (i.e., IRE is much higher than others.), differ in their properties (i.e., IRE is usually used when RFA and MWA are considered no indication.) and different healthcare settings (i.e., IRE has been lacking insurance coverage in our country.). Accordingly, the selection of ablation modality has not been evidence-based but is based on the clinician’s experience. This single-center retrospective study aimed to compare the therapeutic and safety outcomes of RFA, MWA, and IRE in HCC patients.

## 2. Materials and Methods

### 2.1. Patients

This study was approved by the ethics committee of our institution. Owing to the retrospective nature of this study, the need to obtain written informed consent was waived. Patients were identified using an institutional database that tracked HCC patients treated with image-guided ablation. RFA for treatment of HCC became available in 2000, IRE in 2014, and MWA in 2018. All three modalities have become available since then. 

Between January 2018 and October 2021, we identified patients with HCCs who underwent either RFA or MWA as a curative therapy. In addition, between January 2014 and October 2021, we identified patients with HCCs who underwent IRE as a curative therapy. Thus, 322 consecutive patients with 366 HCCs who underwent ultrasound (US)-guided RFA (n = 216, 59.0%), MWA (n = 91, 28.3%), and IRE (n = 15, 4.7%) were included. Patient and tumor characteristics according to the three different ablation therapies are summarized in Table 1 and Table 2, respectively. The imbalance in number of patients who underwent IRE was because IRE has been lacking insurance coverage in our country. Mean tumor size (± standard deviation) was 1.5 ± 0.7 cm for the RFA group, 2.1 ± 1.0 cm for MWA group, and 1.4 ± 0.5 cm for the IRE group (*p* < 0.001). Number of nodules each patient had (1/2/3) was 180/30/6 in RFA, 78/8/5 in MWA, and 8/7/0 in IRE, respectively (*p* = 0.002).

All patients were discussed at our department before treatment. Lesions in direct contact with heat-sensitive structures, such as the major portal vein, were considered unsuitable for thermal ablation and were considered for IRE. The choice between RFA and MWA was mainly based on operator preferences. 

HCC was diagnosed based on typical findings of HCC on contrast-enhanced computed tomography (CECT), gadoxetic acid (Primovist; Bayer Health Care, Osaka, Japan)-enhanced magnetic resonance imaging (EOB-MRI), and/or contrast-enhanced US (CEUS) with a perflubutane microbubble contrast agent (Sonazoid; GE Healthcare, Oslo, Norway), using the non-invasive criteria recommended by the Japan Society of Hepatology [2].

### 2.2. Devices and Ancillary Procedures

All RFAs were performed using 17-gauge, internally cooled electrode applicator (Cool-tip RF ablation System E Series; Medtronic, Minneapolis, MN, USA). MWAs were performed using a 13-gauge antenna of the 2.4-GHz system (Emprint Ablation System; Medtronic, Minneapolis, MN, USA). IREs were performed using two–four 19-gauge monopolar needle electrodes (NanoKnife; AngioDynamics, Latham, NY, USA). All interventions were performed percutaneously using a dedicated US system (Aplio 500 or Aplio i800; Canon Medical Systems, Otawara, Tochigi, Japan) equipped with a 3.75-MHz convex transducer (PVT-385BT or PVI-482BX). 

To identify tumors and ensure precise needle placement, CEUS, CT/MRI/US fusion (Smart Fusion; Canon Medical Systems), and a needle-tracking system (Smart Navigation; Canon Medical Systems) were used. CEUS and CT/MRI/US fusion were most frequently used in IRE (100% [15/15]), and needle tracking was most frequently used in MWA (38.5% [35/91]) (Table 3). 

Ancillary procedures, such as artificial pleural effusion or artificial ascites, were used in selected cases to detect tumors that were adjacent to the diaphragm or were not clearly seen because of the lung artifact and to minimize the risk of thermal injury to adjacent anatomical structures. Artificial ascites was most frequently used in MWA (29.7% [27/91]) (Table 3).

### 2.3. Assessment of Therapeutic Outcomes and Safety Profile

Treatment efficacy was evaluated using CECT or EOB-MRI–1-3 days after ablation using the modified Response Evaluation Criteria in Solid Tumors (mRECIST) [8]. When thermal ablations, such as RFA and MWA, were performed, complete ablation was defined as no tumor enhancement with the creation of a circumferential ablative margin of at least 5 mm. If the safety margin was judged to be insufficient, ablation was performed within 1 month. However, if the circumferential ablative margin was difficult to achieve because of the location of the tumor (i.e., adjacent to the blood vessels), ablation was not permissible. 

Complete ablation for IRE was defined as no tumor enhancement regardless of whether a sufficient safety margin was obtained [9,10]. In cases of incomplete ablation (i.e., intratumoral enhancement), another session of IRE was performed to achieve complete ablation on the same day. Immediately after the IRE procedure, all patients underwent CEUS. One to three days after IRE, CECT, or EOB-MRI was performed to assess the area of tissue ablation. The devascularized area was considered to represent the necrotic area.

Safety parameters were assessed based on immediate and 30-day complications according to the Clavien–Dindo classification system and Society of Interventional Radiology guidelines [11,12].

Follow-up surveillance CECT or EOB-MRI and blood tests, including those for tumor markers such as α-fetoprotein (AFP), AFP-L3, and des-γ-carboxy prothrombin, were performed at intervals of 3 to 4 months. If at least one of the tumor markers was elevated, an additional CECT or EOB-MRI was performed. Patients were followed-up until the date of the last follow-up or death. The median follow-up period per patient was 23.5 months (interquartile range, 13.0–36.0 months). Intrahepatic HCC recurrence was classified as either tumor recurrence at a site distant from the primary tumor or recurrence adjacent to (in contact with) the treated site (LTP). Radiological interpretation of recurrence was performed by a radiologist specializing in body imaging, who was not blinded to the treatment. 

### 2.4. Statistical Analysis

Continuous variables were expressed as mean ± standard deviation or median (interquartile range) and were analyzed using one-way analysis of variance or the Kruskal–Wallis test. Categorical variables were presented as numbers (percentages) and were analyzed using Pearson’s χ^2^ test or Fisher’s exact test. LTP and recurrence–free survival (RFS) probabilities were estimated using the Kaplan–Meier method and compared using the log-rank test. Variables associated with LTP and RFS were assessed using a Cox proportional hazards model. 

Propensity score-matched analysis was performed to reduce the selection bias on LTP analyses by creating matched groups of patients who underwent each modality (RFA vs. MWA, RFA vs. IRE, and MWA vs. IRE). Propensity score-matched analysis using multinomial logistic regression for comparing three arms was not employed because the number of patients of IRE was small. The propensity score model comprised maximum tumor diameter, transcatheter arterial chemoembolization (TACE) prior to ablation, tumor form (simple nodular type or others), and tumor locations (adjacent to (< 3 mm) liver surface [hump], portal vein [major branch], portal vein [minor branch], hepatic duct, hepatic vein, inferior vena cava [IVC], gall bladder, or diaphragm). Thus, the portal vein was classified into two groups: major branch and minor branch. The major branch included the main trunk and first- or second-order branches of the portal vein, and the minor branches were vessels distal to the third-order branch of the portal vein [13]. Propensity scores were calculated by applying these variables to a logistic regression model. One-to-one propensity matching was used to match the cohorts with a nearest-neighbor matching within a caliper width equal to 0.2 of the standard deviation of the propensity score logit. 

Statistical significance was set at *p* < 0.05. All statistical analyses were performed using the JMP software (version 14.0; SAS Institute, Tokyo, Japan) and EZR version 1.55 (Saitama Medical Center, Jichi Medical University, Saitama, Japan).

## 3. Results

### 3.1. Comparisons of LTP Rates between Each Modality Group

Figure 1a shows LTP curves between RFA and MWA in the unmatched cohort. There was no significant difference between the cohorts (*p* = 0.185), and the 1-, 2-, and 3-year LTP rates were 18.8%, 24.3%, and 25.3%, respectively, for RFA and 12.1%, 20.5%, and 20.5%, respectively, for MWA. Figure 1b shows LTP curves between RFA and MWA in the propensity score-matched cohort. There was no significant difference between the cohorts (*p* = 0.586), and the 1-, 2-, and 3-year LTP rates were 16.6%, 18.2%, and 18.2%, respectively, for RFA and 12.1%, 20.6%, and 20.6%, respectively, for MWA.

Figure 1c shows LTP curves between RFA and IRE in the unmatched cohort. There were significant differences between the cohorts (*p* = 0.028), and the 1-, 2-, and 3-year LTP rates were 18.8%, 24.5%, and 25.3%, respectively, for RFA and 0.0%, 0.0%, and 0.0%, respectively, for IRE. Figure 1d shows LTP curves between RFA and IRE in the propensity score-matched cohort. There was a significant difference between the cohorts (*p* = 0.005), and the 1-, 2-, and 3-year LTP rates were 31.3%, 45.0%, and not available, respectively, for RFA and 0.0%, 0.0%, and 0.0%, respectively, for IRE.

Figure 1e shows LTP curves between MWA and IRE in the unmatched cohort. There was no significant difference between the cohorts (*p* = 0.083), and the 1-, 2-, and 3-year LTP rates were 12.1%, 20.5%, and 20.5%, respectively, for MWA and 0.0%, 0.0%, and 0.0%, respectively, for IRE. Figure 1f shows LTP curves between MWA and IRE in the propensity score-matched cohort. There was no significant difference between the cohorts (*p* = 0.103), and the 1-, 2-, and 3-year LTP rates were 10.0%, 25.0%, and 25.0%, respectively, for MWA and 0.0%, 0.0%, and 0.0%, respectively, for IRE.

### 3.2. Factors Contributing to LTP of Each Modality

Regarding predisposing factors for LTP after RFA, the univariate analysis revealed that maximum diameter, TACE prior to RFA, tumor form (simple nodular or others), tumor location (portal vein [major branch] and IVC), and ablated margin (<3 mm) were significant predisposing factors for LTP. Multivariate analysis revealed that tumor location (IVC) (hazard ratio [HR]: 3.239,95% confidence interval [CI]: 1.114–9.423) (*p* = 0.031) and ablation margin (<3 mm) (HR: 3.982, 95% CI: 2.077–7.633) (*p* < 0.001) were independent predisposing factors for LTP (Table 4). Some tumor location factors such as gall bladder, colon, heart, and stomach were not estimated definitely due to imbalance of the number.

Regarding predisposing factors for LTP after MWA, the univariate analysis revealed that maximum diameter, TACE prior to MWA, tumor form (simple nodular or others), tumor location (IVC), and ablated margin (<3 mm) were significant predisposing factors for LTP. Multivariate analysis revealed that only ablation margin (<3 mm) (HR: 3.982, 95% CI: 2.077–7.633) (*p* < 0.001) was an independent predisposing factor for LTP (Table 5). Some tumor location factors such as colon, heart, stomach, and kidney were not estimated definitely due to imbalance of the number.

### 3.3. Factors Contributing to RFS after Ablation

Regarding predisposing factors for RFS after ablation, the univariate analysis revealed that etiology (hepatitis B virus and nonviral), Child-Pugh score, platelet count, and AFP-L3 were significant predisposing factors for RFS. Multivariate analysis revealed that only AFP-L3 (HR: 1.013, 95% CI: 1.005–1.021) (*p* = 0.002) was an independent predisposing factor for RFS (Table 6).

### 3.4. Comparison of Complication Each Modality Group

A comparison of the complication profiles in each ablation group is shown in Table 7. There was no significant difference in grade I (*p* = 0.050) and grade II–V (*p* = 0.455) complications between the groups. However, one patient death occurred within 30 days after MWA due to exacerbation of interstitial pneumonia, the cause of which may be artificial pleural effusion during the ablation procedure.

## 4. Discussion

A direct comparison of the clinical outcomes of different ablative modalities such as RFA, MWA, and IRE was conducted. The study revealed that IRE showed significantly better local tumor control than RFA in both unmatched (*p* = 0.028) and propensity score-matched cohorts (*p* = 0.005). In this study, 33.3% of tumors treated with IRE were located adjacent to major portal branches, which was statistically more frequent than others, and no LTP was observed in patients treated with IRE. A possible reason for the unfavorable effect of RFA on perivascular tumors is that the heat-sink effect may have caused insufficient ablation and LTP [14]. Thus, IRE rather than RFA should be used to treat tumors adjacent to the vascular structure to prevent LTP.

Owing to the nature of IRE, which requires multi-needle insertion and general anesthesia and is relatively expensive compared with RFA and MWA, high-volume prospective registration and randomized controlled trials that directly address the added value of IRE over other modalities, such as RFA and MWA, are difficult to perform. Based on the cumulative clinical IRE literatures, which are largely retrospective reports and prospective phase I or II trials that use different inclusion criteria and outcome measures, although clinical results are largely promising [9,15,16,17,18,19], IRE was associated with inferior local tumor control [20], which is different from our study results.

Here, we address some possible reasons for these findings. First, the size of the tumor treated with IRE in this study was relatively small compared with that in other studies [15,16,17,18,19]. Specifically, median tumor size treated with IRE in our study was 1.3 cm (range, 0.7 to 2.3 cm). In contrast, in previous studies median tumor sizes treated with IRE ranged from 1.9 to 2.6 cm [14,15,16,17,18]. Second, we used the CT/MRI fusion system in all cases treated with IRE, which can display the “C-plane” (the plane perpendicular to the electrode line) in addition to the “B-plane” (the normal US plane) (Figure 2), which may facilitate the accurate deployment of multiple electrodes around or inside the tumors [21]. Finally, we performed CEUS immediately after IRE ablation in all cases. Our treatment endpoint for IRE was to observe a loss of intratumoral enhancement, and if we detected partial intratumoral enhancement, we performed additional ablation [9,10,21]. 

In contrast to RFA, there was no significant difference in local tumor control between IRE and MWA in either the unmatched (*p* = 0.083) or propensity score-matched cohorts (*p* = 0.103). In addition, regarding factors potentially associated with LTP in RFA, small ablative margin (i.e., <3 mm) and tumor location (adjacent to IVC) were the significant factors in multivariate analysis. In contrast, with respect to MWA, a small ablative margin was the only significant factor associated with LTP in the multivariate analysis. A possible reason for this is that the heat-sink effect may have a greater influence on MWA than on RFA. Although the technical characteristics of MWA and RFA are similar, they exhibit several differences in their physical mechanisms of thermogenesis [22,23]. The significant difference is that during MWA, heat is generated in a fixed space around the antenna applicator, whereas during RFA, heat is confined to zones of high current density. Although some studies have demonstrated the utility of MWA over RFA for perivascular tumors [5,14], further studies are needed to clarify whether MWA can reduce the risk of LTP for perivascular HCCs without injuring Glisson’s sheath.

As mentioned, although MWA may have the same advantages over RFA, the same disadvantages exist: one is that the needle tip is difficult to see by US, especially for deep-seated lesions. To overcome this, we used a virtual needle tracking system that tracks the position of the needle tip using a small sensor on the shaft, which could be a helpful method for achieving more precise monitoring of MWA needle tips during puncture and ablation. Second, the MWA needle (13-gauge) was thicker than the others (17-gauge for RFA and 19-gauge for IRE), which may cause bleeding. However, there was no difference between MWA and RFA in terms of the frequency and severity of the complications.

In the present study, no significant difference in 2-year RFS was observed among the three modalities (RFA, MWA, and IRE: 39.3%, 44.7%, and 52.9%, respectively; *p* = 0.226). Regarding factors potentially associated with RFS after ablation, AFP-L3 was the only significant factor in the multivariate analysis because distance recurrence depends mainly on the carcinogenic potential of non-cancerous tissues. Recently, preclinical studies have revealed that ablation therapy can play a considerable role in distant lesions through immune effects, known as the abscopal effect [24,25,26]. However, we did not observe any abscopal effects in our study. Combination strategies of immunotherapy and ablation may be promising in the future [27].

Our study had several limitations. First, this was a retrospective study, and prospective randomized study could be warranted. Second, there was an extreme imbalance in the sample size among ablation therapies, which may have introduced bias and limited our ability to arrive at definitive conclusions. Third, although we adopted propensity score-matched analysis to reduce bias, potential selection and indication biases are inevitable due to the retrospective nature of the study. Finally, differing levels of operator experience with ablation may have affected the outcomes.

## 5. Conclusions

In conclusion, IRE provides better local tumor control than RFA as a first-line therapeutic option for small perivascular HCC. RFA and MWA offer similar therapeutic and safety outcomes in patients with early stage HCC. Prospective randomized trials are warranted to establish an evidence basis for ablative modality selection for the treatment of early stage HCC. 

## Figures and Tables

**Figure 1 cancers-15-00732-f001:**
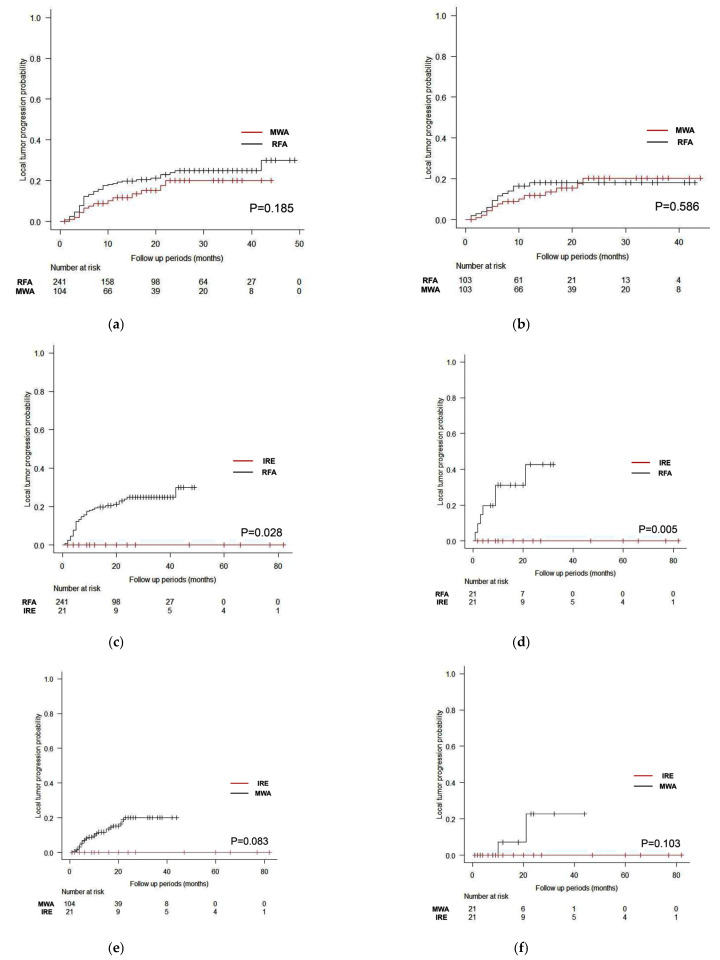
Local tumor progression (LTP) curves with log-rank test stratified by treatment modality with and without propensity score matching. Regarding radiofrequency ablation (RFA) and microwave ablation (MWA), there was no significant difference between cohorts, both with (**a**) (p = 0.185) and without (**b**) (*p* = 0.586) propensity score matching. Regarding RFA and irreversible electroporation (IRE), there was a significant difference between cohorts, both with (**c**) (*p* = 0.028) and without (**d**) (*p* = 0.005) propensity score matching. Regarding MWA and IRE, there was no significant difference between cohorts, both with (**e**) (*p* = 0.083) and without (**f**) (*p* = 0.103) propensity score matching.

**Figure 2 cancers-15-00732-f002:**
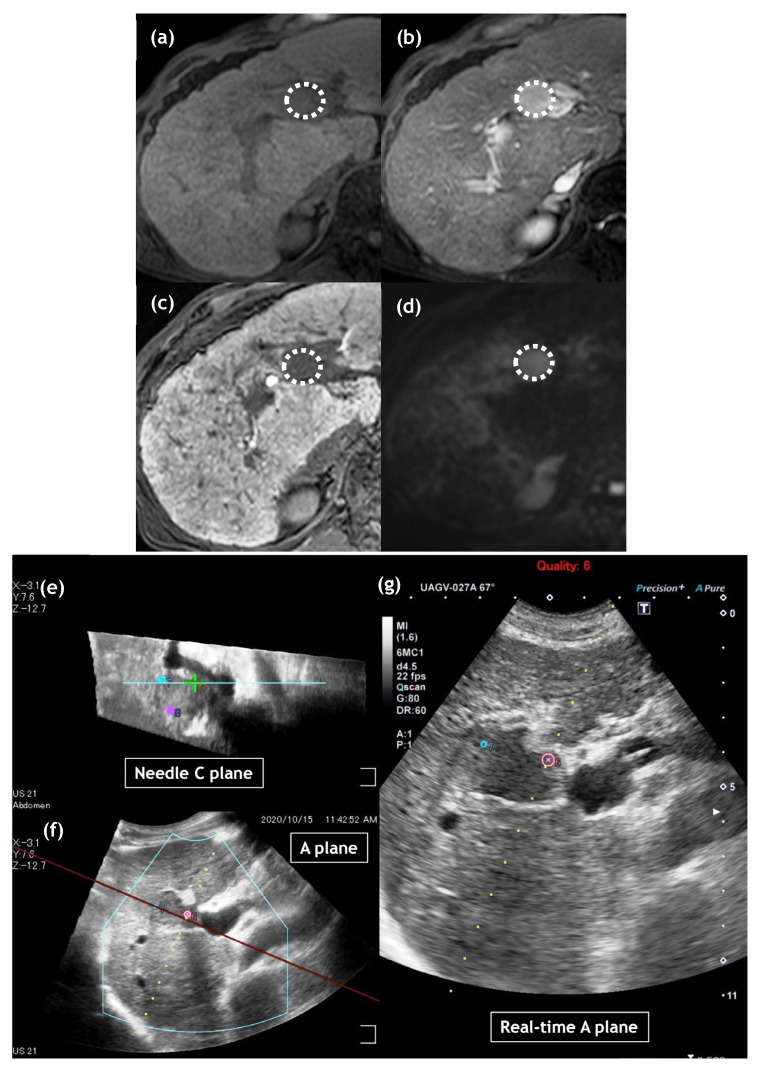
Images of a 70-year-old man with hepatocellular carcinoma (HCC) of 23 mm diameter in liver segment 4. On gadoxetic acid-enhanced magnetic resonance imaging, HCC shows hypointense in the pre-contrast T1-weighted image (**a**: dotted circle), hyperintensity in the arterial phase (**b**: dotted circle) and hypointensity in the hepatobiliary phase (**c**: dotted circle). Diffusion-weighted image shows diffusion restriction (**d**: dotted circle). Multiple ultrasound (US) images were reconstructed from US volume data collected using a magnetic sensor. The upper left image shows the reconstructed needle C-plane, which is perpendicular to the needle line, including the center of the tumor (**e**). The under left image shows the reconstructed A-plane, which is a normal US view (**f**). The image on the right is the real-time US image (**g**). On US-guided irreversible electroporation (IRE) planning, three spherical markers were placed on the needle C-plane (upper left: **e**)**,** each distance of which was 18 mm. Each marker demonstrates a virtual IRE needle position, which helps us deploy IRE needles during treatment.

**Table 1 cancers-15-00732-t001:** Patient characteristics according to three different ablation therapies.

Patient Basis Characteristics	RFA (n = 216)	MWA (n = 91)	IRE (n = 15)	*p*-Value
Sex (Male/female)	160/56	72/19	12/3	0.594
Mean age (years)	73.8 ± 8.7	72.3 ± 10.5	73.8 ± 5.0	0.620
Etiology (HCV/HBV/HCV+HBV/other)	87/36/1/92	37/7/1/46	5/1/0/9	0.357
Naïve/non-naive	55/161	37/54	3/12	**0.020**
Child-Pugh score (5/6/7/8/9)	173/33/6/3/1	72/15/2/1/1	11/3/0/0/1	0.552
T-Bil (mg/dL)	0.64 [0.52, 0.87]	0.67 [0.51, 0.98]	0.53 [0.43, 0.83]	0.742
Alb (g/dL)	3.8 [3.5, 4.1]	3.9 [3.5, 4.2]	4.2 [3.4, 4.4]	0.122
PT-INR	1.04 [0.98, 1.11]	1.05 [0.99, 1.14]	1.01 [1.00, 1.08]	0.410
Plt (×10^4^)	13.5 [10.1, 16.5]	14.6 [11.1, 18.4]	14.4 [8.0, 15.7]	0.249
AFP (ng/mL)	5.4 [2.7, 13.9]	5.4 [2.6, 10.2]	10.7 [4.2, 19.8]	0.238
AFP-L3 (%)	0.5 [0.5, 8.7]	0.5 [0.5, 7.2]	0.5 [0.5, 7.8]	0.771
DCP (mAU/mL)	27 [18, 66]	34 [22, 89]	34 [20, 135]	0.153
Number of nodules (1/2/3)	180/30/6	78/8/5	8/7/0	**0.002**
Maximum tumor diameter (cm)	1.5 ± 0.7	2.1 ± 1.0	1.4 ± 0.5	**<0.001**
TACE before ablation (%)	4.2% (9/216)	11.0% (10/91)	0% (0/15)	**0.042**

**Note.** Variables are expressed as mean ± standard deviation or median [interquartile range]. Abbreviations: RFA, radiofrequency ablation; MWA, microwave ablation; IRE, irreversible electroporation; HCV, hepatitis virus C; HBV, hepatitis B virus; T-Bil, total bilirubin; PT-INR, prothrombin time-international normalized ratio; Plt, platelet; AFP, α-fetoprotein; DCP, des-γ-carboxy prothrombin; TACE, transcatheter arterial chemoembolization.

**Table 2 cancers-15-00732-t002:** Tumor characteristics according to three different ablation therapies.

Tumor Basis Characteristics	RFA (n = 241)	MWA (n = 104)	IRE (n = 21)	*p*-Value
Maximum tumor diameter (cm)	1.5 ± 0.7	2.1 ± 1.0	1.4 ± 0.5	**<0.001**
Tumor form (Simple nodular type/others)	185/56	73/30	21/0	**0.016**
Couinaud classification; S1/S2/S3/S4/S5/S6/S7/S8	18/24/19/22/27/41/33/57	2/6/7/11/6/28/13/31	2/1/1/2/4/5/4/2	0.215
Tumor location (adjacent to; )				
Liver surface (hump) (%)	35.3% (85/241)	48.1% (50/104)	0% (0/21)	**<0.001**
Portal vein (major branch) (%)	15.8% (38/241)	4.8% (5/104)	33.3% (7/21)	**<0.001**
Portal vein (minor branch) (%)	20.3% (49/241)	24.0% (25/104)	14.3% (3/21)	0.545
Hepatic duct (%)	1.7% (4/241)	1.0% (1/104)	0% (0/21)	0.752
Hepatic vein (%)	14.9% (36/241)	10.6% (11/104)	4.8% (1/21)	0.276
IVC (%)	2.5% (6/241)	1.9% (2/104)	0% (0/21)	0.738
Gall bladder (%)	2.1% (5/241)	2.9% (3/104)	9.5% (2/21)	0.132
Colon (%)	2.5% (6/241)	1.9% (2/104)	9.5% (2/21)	0.139
Heart (%)	0.4% (1/241)	1.9% (2/104)	0% (0/21)	0.330
Stomach (%)	0.8% (2/241)	1.9% (2/104)	4.8 (1/21)	0.279
Duodenum (%)	0% (0/241)	0% (0/104)	4.8% (1/21)	**<0.001**
Diaphragm (%)	11.2% (27/241)	18.3% (19/104)	0% (0/21)	**0.039**
Kidney (%)	1.2% (3/241)	1.9% (2/104)	0% (0/21)	0.757

**Note.** Variables are expressed as mean ± standard deviation. Abbreviations: RFA, radiofrequency ablation; MWA, microwave ablation; IRE, irreversible electroporation; IVC, inferior vena cava.

**Table 3 cancers-15-00732-t003:** Ancillary procedures according to three different ablation therapies.

Characteristics	RFA (n = 216)	MWA (n = 91)	IRE (n = 15)	*p*-Value
Treatment support system				
CEUS (%)	49.5% (107/216)	50.6% (46/91)	100% (15/15)	**<0.001**
CT/MRI/US fusion (%)	55.6% (120/216)	40.7% (37/91)	100% (15/15)	**<0.001**
Needle tracking (%)	2.8% (6/216)	38.5% (35/91)	26.7% (4/15)	**<0.001**
Ancillary procedures				
Artificial pleural effusion (%)	12.0% (26/216)	9.9% (9/91)	6.7% (1/15)	0.734
Artificial ascites (%)	24.1% (52/216)	29.7% (27/91)	0% (0/15)	**0.045**

**Note.** Abbreviations: RFA, radiofrequency ablation; MWA, microwave ablation; IRE, irreversible electroporation; CEUS, contrast-enhanced ultrasound; CT, computed tomography; MRI, magnetic resonance imaging.

**Table 4 cancers-15-00732-t004:** Predisposing factors for local tumor progression after RFA.

	Hazard Ratio (95% CI)				
Variables		Univariate	*p*-Value	Multivariate	*p*-Value
Number of tumors	241				
Maximum diameter		1.722 (1.195–2.482)	**0.004**	1.106 (0.7191–1.701)	0.647
TACE prior to RFA		3.953 (1.682–9.294)	**0.002**	1.641 (0.628–4.286)	0.312
Tumor form (Simple nodular type/others)		1.906 (1.051–3.457)	**0.034**	1.392 (0.736–2.632)	0.306
Tumor location (adjacent to; )					
Liver surface (hump)		0.908 (0.501–1.647)	0.751		
Portal vein (major branch)		3.102 (1.740–5.531)	**<0.001**	1.821 (0.962–3.446)	0.066
Portal vein (minor branch)		0.5166 (0.220–1.214)	0.130		
Hepatic duct		1.777 (0.430–7.342)	0.427		
Hepatic vein		1.184 (0.575–2.436)	0.646		
IVC		3.79 (1.359–10.570)	**0.011**	3.239 (1.114–9.423)	**0.031**
Gall bladder		0.0000001085 (0-Inf)	0.996		
Colon		0.0000001079 (0-Inf)	0.995		
Heart		0.0000003033 (0-Inf)	0.997		
Stomach		0.0000006247 (0-Inf)	0.996		
Duodenum		1 (1–1)	NA		
Diaphragm		0.677 (0.243–1.883)	0.455		
Kidney		1.721 (0.237–12.490)	0.591		
Ablated margin <3 mm		5.807 (3.244–10.400)	**<0.001**	3.982 (2.077–7.633)	**<0.001**

**Note.** Abbreviations: CI, confidence interval; TACE, transcatheter arterial chemoembolization; RFA, radiofrequency ablation; IVC, inferior vena cava; NA, not available; Inf, infinite.

**Table 5 cancers-15-00732-t005:** Predisposing factors for local tumor progression after MWA.

	Hazard Ratio (95% CI)				
Variables		Univariate	*p*-Value	Multivariate	*p*-Value
Number of tumors	104				
Maximum diameter		1.577 (1.060–2.347)	**0.025**	1.064 (0.624–1.814)	0.819
TACE prior to MWA		3.34 (1.118–9.981)	**0.031**	1.630 (0.446–5.957)	0.460
Tumor form (Simple nodular type/others)		3.529 (1.223–10.180)	**0.020**	1.414 (0.423–4.728)	0.573
Tumor location (adjacent to; )					
Liver surface (hump)		2.203 (0.738–6.581)	0.157		
Portal vein (major branch)		1.353 (0.175–10.430)	0.772		
Portal vein (minor branch)		0.9884 (0.310–3.155)	0.984		
Hepatic duct		7.376 (0.940–57.860)	0.057		
Hepatic vein		2.141 (0.596–7.697)	0.243		
IVC (%)		10.64 (1.229–92.190)	**0.032**	5.332 (0.474–59.940)	0.175
Gall bladder		1.774 (0.231–13.610)	0.582		
Colon		0.0000001086 (0-Inf)	0.998		
Heart		0.0000001086 (0-Inf)	0.998		
Stomach		0.0000001102 (0-Inf)	0.999		
Duodenum		1 (1–1)	NA		
Diaphragm		2.020 (0.633–6.453)	0.235		
Kidney		0.0000001086 (0–Inf)	0.998		
Ablated margin <3 mm		36.360 (5.002–294.200)	**<0.001**	31.3 (3.95–248.1)	**0.001**

**Note.** Abbreviations: CI, confidence interval; TACE, transcatheter arterial chemoembolization; MWA, microwave ablation; IVC, inferior vena cava; NA, not available; Inf, infinite.

**Table 6 cancers-15-00732-t006:** Factors predisposing patients to recurrence of HCC after ablation.

	Hazard ratio (95% CI)				
Variables		Univariate	*p*-Value	Multivariate	*p*-Value
Number of patients					
Age		1.002 (0.986–1.018)	0.855		
Gender		1.319 (0.918–1.893)	0.134		
Etiology (HCV)		0.895 (0.663–1.209)	0.470		
Etiology (HBV)		0.619 (0.396–0.969)	**0.036**	0.763 (0.470–1.238)	0.273
Etiology (nonviral)		1.411 (1.052–1.892)	**0.022**	1.213 (0.882–1.667)	0.235
Child-Pugh score		1.408 (1.100–1.802)	**0.007**	1.184 (0.913–1.536)	0.203
Platelet count		0.971 (0.948–0.995)	**0.018**	0.981 (0.957–1.005)	0.112
AST		1 (0.995–1.006)	0.894		
ALT		1.001 (0.996–1.006)	0.696		
AFP		0.100 (0.999–1)	0.349		
AFP-L3		1.016 (1.008–1.024)	**<0.001**	1.013 (1.005–1.021)	**0.002**
DCP		1 (1–1)	0.205		
Tumor number		0.988 (0.823–1.185)	0.893		
Ablation method (RFA)		1.241 (0.895–1.721)	0.195		
Ablation method (MWA)		0.908 (0.646–1.276)	0.576		
Ablation method (IRE)		0.534 (0.236–1.210)	0.133		
Naïve or not		0.728 (0.520–1.021)	0.066		

**Note.** Abbreviations: RFA, radiofrequency ablation; MWA, microwave ablation; IRE, irreversible electroporation; HCV, hepatitis virus C; HBV, hepatitis B virus; T-Bil, total bilirubin; PT-INR, prothrombin time-international normalized ratio; Plt, platelet count; AST, aspartate aminotransferase; ALT, alanine aminotransferase; AFP, α-fetoprotein; DCP, des-γ-carboxy prothrombin.

**Table 7 cancers-15-00732-t007:** Complications based on the Clavien–Dindo classification.

Modality	Total (n = 322)	RFA (n = 216)	MWA (n = 91)	IRE (n = 15)	*p*-Value
Grade I	17/322 (5.28%)	7/216 (3.24%)	8/91 (8.79%)	2/15 (13.3%)	0.050
Grade II–V	6/322 (1.86%)	3/216 (1.39%)	3/91 (3.30%)	0/15 (0.00%)	0.455
Details of Grade II ≥					
Pleural effusion	0	0	1 (III)	0	NA
Pneumothorax	0	1 (III)	0	0	NA
Interstitial pneumonia	0	0	1 (V)	0	NA
Liver abscess	0	2 (II)	1 (II)	0	NA

**Note.** Abbreviations: RFA, radiofrequency ablation; MWA, microwave ablation; IRE, irreversible electroporation; NA, not available.

## Data Availability

The data that support the findings of this study are available on request form the corresponding author. The data are not publicly available due to privacy or ethical restrictions.

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
