# Peer review of "Comparisons of Radiofrequency Ablation, Microwave Ablation, and Irreversible Electroporation by Using Propensity Score Analysis for Early Stage Hepatocellular Carcinoma"

_cancers, 2023, doi:10.3390/cancers15030732_

Round 1

Reviewer 1 Report

Dear Authors

I have appreciated your interesting and complete paper.

It is easy to read although rich in information, and you have considered many variables. The relative small number of patients treated with IRE is certainly a limitation but overall numbers are significant.

There are only a little number of comments and requests of revision:

1.       In Assessment of Therapeutic Outcomes it is better to specify which criteria has been used to assess recurrence (RECIST, m-RECIST, LI-RADS or others)

2.       Figure 2 caption is a little confused in reading and to me it is not clear the difference in the message conveyed by (e) and (f) figures

3.       A comprehensive legend of all the acronyms used could be useful

Author Response

Reviewer1

Thank you very much for your helpful and constructive comments on our paper (cancers-2121610) entitled "Comparisons of Radiofrequency Ablation, Microwave Ablation, and Irreversible Electroporation by Using Propensity Score Analysis for Hepatocellular Carcinoma". We have revised our manuscript based on all of the reviewers’ comments.

Please see the annotated revised manuscript, to which we have made changes to address the reviewers’ comments. Also, our detailed responses to each of your comments are given below.

I have appreciated your interesting and complete paper.

It is easy to read although rich in information, and you have considered many variables. The relative small number of patients treated with IRE is certainly a limitation but overall numbers are significant.

There are only a little number of comments and requests of revision:

R1-1. In Assessment of Therapeutic Outcomes it is better to specify which criteria has been used to assess recurrence (RECIST, m-RECIST, LI-RADS or others)

Reply: In this study we used mRECIST for the assessment of treatment outcomes. Thus we have added it in the text and also cited in ref.#8.

R2-1. Figure 2 caption is a little confused in reading and to me it is not clear the difference in the message conveyed by (e) and (f) figures

Reply: Thank you for your suggestion. As you pointed out, the difference of Fig. e and f is very small. Thus, we have delated Fig. e. Also. We have also corrected some grammatical mistakes.

R1-3. A comprehensive legend of all the acronyms used could be useful

Reply: Thank you for your comment.

Reviewer 2 Report

The authors present the outcomes from a retrospective single-center series aiming to evaluate therapeutic and safety outcomes from different ablation techniques in patients with small HCC. This is an interesting study on a specific patient population with small HCC. It is well-structured and written. A number of issues to be addressed by the authors:

- Reorder methods and results. All tables and relative outcomes should be moved to the outcomes section strictly.

- The limitations section needs to be improved ie. retrospective nature etc.

- I would propose adding "Early" or "Small" HCC in the title to be more precise based on the outcomes presented.

Author Response

Reviewer2

Thank you very much for your helpful and constructive comments on our paper (cancers-2121610) entitled "Comparisons of Radiofrequency Ablation, Microwave Ablation, and Irreversible Electroporation by Using Propensity Score Analysis for Hepatocellular Carcinoma". We have revised our manuscript based on all of the reviewers’ comments.

Please see the annotated revised manuscript, to which we have made changes to address the reviewers’ comments. Also, our detailed responses to each of your comments are given below.

The authors present the outcomes from a retrospective single-center series aiming to evaluate therapeutic and safety outcomes from different ablation techniques in patients with small HCC. This is an interesting study on a specific patient population with small HCC. It is well-structured and written. A number of issues to be addressed by the authors:

R2-1. Reorder methods and results. All tables and relative outcomes should be moved to the outcomes section strictly.

Reply: Thank you for your suggestion. This study is not a prospective but retrospective study. Thus we still think there is no need to place all the Tables’ information on “results section”.

R2-2. The limitations section needs to be improved ie. retrospective nature etc.

Reply: Thank you for your suggestion. We have added it to the limitation section.

R2-3. I would propose adding "Early" or "Small" HCC in the title to be more precise based on the outcomes presented.

Reply: Thank you for your comment. We have added “early stage” to the title.

Reviewer 3 Report

This article describes a clinical retrospective based on comparing the clinical outcomes after ablations on early stage hepatocellular carcinoma (HCC) with three electromagnetic energies: RF, MW and high-voltage pulses (IRE). The article is well written and the methodology seems to be correct too. My main concern is about the clinical impact, and principally the clinical evidence level and quality. The authors must establish the level and quality of clinical evidence offered by their data and analysis. This is closely related to how the authors state the conclusions and describe the main findings in the abstract. At this regard, there are some points to be clarified:

1) In the abstract is said that “A significant difference in 2-year LTP rates between the IRE and RFA groups (IRE, 0.0% vs. RFA, 45.0%; p = .005) was found”. However, in the Discussion section is said that “no significant difference in 2-year RFS was observed among the three modalities (RFA, MWA, and IRE: 39.3%, 44.7%, and 52.9%, respectively; p = .226). These results seem not to be matched.

2) My personal feeling is that IRE group could have been in a way ‘favorably’ treated since 1) CT/MRI fusion system was used to facilitate the accurate deployment of multiple electrodes around or inside the tumors, and 2) additional ablations were conducted if partial intra-tumoral enhancement was detected just after the first one.

3) Needle-tracking system to identify tumors and ensure precise needle placement were hardly used in RFA and less than the half of the MWA cases. How this could impact the results?

4) Information about RFS should be also included in the Conclusions section since it’s very relevant.

Major comments:

1) L63: Consider extending the introduction by comment the scarce literature about cross-comparisons of RFA, MWA and IRE.

2) L308: It is said that mean tumor size in the IRE Group is 1.3 cm, which is not the same as shown in Table 2 (1.4 ± 0.5 cm). If this is due to the different used metric (mean vs. maximum value). Clarify this point.

Minor comments:

L51: MWA instead of MVA

L81-82: Consider including here the statistical significance associated with the lesion size and number of nodules (as indicated in Table 1).

Table 1: Consider pointing out in bold those values with statistical significance (P<0.05).

L255: MWA instead of RFA

L308: Change (14-18) to [14-18]

Author Response

Reviewer3

Thank you very much for your helpful and constructive comments on our paper (cancers-2121610) entitled "Comparisons of Radiofrequency Ablation, Microwave Ablation, and Irreversible Electroporation by Using Propensity Score Analysis for Hepatocellular Carcinoma". We have revised our manuscript based on all of the editors’ and reviewers’ comments.

Please see the annotated revised manuscript, to which we have made changes to address the reviewers’ comments. Also, our detailed responses to each of your comments are given below.

This article describes a clinical retrospective based on comparing the clinical outcomes after ablations on early stage hepatocellular carcinoma (HCC) with three electromagnetic energies: RF, MW and high-voltage pulses (IRE). The article is well written and the methodology seems to be correct too. My main concern is about the clinical impact, and principally the clinical evidence level and quality. The authors must establish the level and quality of clinical evidence offered by their data and analysis. This is closely related to how the authors state the conclusions and describe the main findings in the abstract. At this regard, there are some points to be clarified:

Reply: Thank you for your important comments

R3-1. In the abstract is said that “A significant difference in 2-year LTP rates between the IRE and RFA groups (IRE, 0.0% vs. RFA, 45.0%; p = .005) was found”. However, in the Discussion section is said that “no significant difference in 2-year RFS was observed among the three modalities (RFA, MWA, and IRE: 39.3%, 44.7%, and 52.9%, respectively; p = .226). These results seem not to be matched.

Reply: Thank you for pointing out. I have made sure the sentence and found that this is correct. You may have misunderstood “recurrence free survival (RFS)” for “local tumor progression (LTP). RFS included not only local tumor progression but also other recurrences such as intrahepatic and extrahepatic recurrences.

R3-2. My personal feeling is that IRE group could have been in a way ‘favorably’ treated since 1) CT/MRI fusion system was used to facilitate the accurate deployment of multiple electrodes around or inside the tumors, and 2) additional ablations were conducted if partial intra-tumoral enhancement was detected just after the first one.

Reply: Thank you for your important opinion. We still think that the tumors treated by IRE were not easy to treat compared to those treated with RFA and MWA. The reasons are as follows: 1) as demonstrated in Table 2, tumors treated by IRE were mainly located in central portion of the liver and 33.3% of which were adjacent to major portal vein. We think these were difficult to treat. In contrast, tumors treated by RFA and MWA were mainly located in peripheral portion of the liver. 2) CT/MRI fusion with needle C-plane enabled us to understand the special relationship between IRE needles and vascular structures. That’s why we used CT/MRI fusion in all IRE cases. In contrast, on RFA and MWA cases, we only used CT MRI fusion when tumors were difficult to detect by ultrasound because needle deployments were easier than that of IRE. 3) In our experience CEUS assessment for residual tumors after thermal ablation is less useful than that after IRE because water vapor arising after thermal ablation hinders CEUS observations. In contrast, tumors after IRE ablation can see more clearly using CEUS. That’s why we performed CEUS after IRE ablation. Accordingly, without CEUS just after thermal ablation, it would have had less impact on LTP than IRE.

R3-3. Needle-tracking system to identify tumors and ensure precise needle placement were hardly used in RFA and less than the half of the MWA cases. How this could impact the results?

Reply: Thank you for your important comment. We think that this could be less impact on the results. Needle-tracking system was used when needle-tip was difficult to detect by ultrasound. As you may know, Emprint’s needle-tip is sometimes difficult to detect compared to RFA and IRE needles. Accordingly Needle-tracking system was used in MWA more often than in RFA and IRE.

R3-4. Information about RFS should be also included in the Conclusions section since it’s very relevant.

Reply: As I stated earlier, RFS included not only local tumor progression but also other recurrences such as intrahepatic and extrahepatic recurrences. In the study we focused on LTP among three modalities. Thus, We still think RFS results should not be included in the conclusion.

Major comments:

R3-5. L63: Consider extending the introduction by comment the scarce literature about cross-comparisons of RFA, MWA and IRE.

Reply: Thank you for your suggestion. We have changed the sentence according to your advice. Please refer to “R3-5” in the annotated copy.

R3-6. L308: It is said that mean tumor size in the IRE Group is 1.3 cm, which is not the same as shown in Table 2 (1.4 ± 0.5 cm). If this is due to the different used metric (mean vs. maximum value). Clarify this point.

Reply: Thank you for pointing out. We have checked this and found correct. In Table 2, the size is expressed as mean± standard deviation. In contrast, in discussion the size is expressed as median and range, which would be easy to understand in the context.

Minor comments:

R3-7. L51: MWA instead of MVA

Reply: Thank you for pointing out. We have corrected it.

R3-8. L81-82: Consider including here the statistical significance associated with the lesion size and number of nodules (as indicated in Table 1).

Reply: Thank you for your suggestion. We have changed this part according to your suggestion. Please refer to “R3-8” in the annotated copy.

R3-9. Table 1: Consider pointing out in bold those values with statistical significance (P<0.05).

Reply: Thank you for your suggestion. We have changed them in bold figures on all the tables.

R3-10. L255: MWA instead of RFA

Reply: Thank you for pointing out. We have corrected it.

R3-11. L308: Change (14-18) to [14-18]

Reply: Thank you for pointing out. We have corrected it.

Round 2

Reviewer 3 Report

The authors have addressed suitably my comments.